# Mimickers of Urothelial Carcinoma and the Approach to Differential Diagnosis

**Claudia Manini** [1]**, Javier C. Angulo** [2,3] **and José I. López** [4,*]

1    Department of Pathology, San Giovanni Bosco Hospital, 10154 Turin, Italy; claudiamaninicm@gmail.com
2    Clinical Department, Faculty of Medical Sciences, European University of Madrid, 28907 Getafe, Spain;
     jangulo1964@gmail.com
3    Department of Urology, University Hospital of Getafe, 28905 Getafe, Spain
4    Department of Pathology, Cruces University Hospital, Biocruces-Bizkaia Health Research Institute,
     48903 Barakaldo, Spain
*    Correspondence: jilpath@gmail.com; Tel.: +34-94-600-6084

**Abstract:** A broad spectrum of lesions, including hyperplastic, metaplastic, inflammatory, infectious, and reactive, may mimic cancer all along the urinary tract. This narrative collects most of them from a clinical and pathologic perspective, offering urologists and general pathologists their most salient definitory features. Together with classical, well-known, entities such as urothelial papillomas (conventional (UP) and inverted (IUP)), nephrogenic adenoma (NA), polypoid cystitis (PC), fibroepithelial polyp (FP), prostatic-type polyp (PP), verumontanum cyst (VC), xanthogranulomatous inflammation (XI), reactive changes secondary to BCG instillations (BCGitis), schistosomiasis (SC), keratinizing desquamative squamous metaplasia (KSM), post-radiation changes (PRC), vaginal-type metaplasia (VM), endocervicosis (EC)/endometriosis (EM) (müllerianosis), malakoplakia (MK), florid von Brunn nest proliferation (VB), cystitis/ureteritis cystica (CC), and glandularis (CG), among others, still other cellular proliferations with concerning histological features and poorly understood etiopathogenesis like IgG4-related disease (IGG4), PEComa (PEC), and pseudosarcomatous myofibroblastic proliferations (post-operative spindle cell nodule (POS), inflammatory myofibroblastic tumor (IMT)), are reviewed. Some of these diagnoses are problematic for urologists, other for pathologists, and still others for both. Interestingly, the right identification of their definitory features will allow their correct diagnoses, thus, avoiding overtreatment. The literature selected for this review also focuses on the immunohistochemical and/or molecular data useful to delineate prognosis.

**Keywords:** urinary tract; pseudotumor; diagnosis; symptoms; histology

## 1. Introduction

Simulators of malignancy are varied and frequently seen in the urinary tract. Benign tumors, as well as infections, reactive inflammations, metaplastic and hyperplastic changes, and other conditions, may eventually mimic malignancy all along the urinary tract, from the renal pelvis to the urethra. Some of them are commonly seen, but others are very uncommon and remain a challenge for the urologist when performing cystoscopies or at the time to make decisions when evaluating radiological images. Some of them remain a challenge for the pathologist who must decide if the submitted material includes any lesion representing one of the many faces of bladder cancer [1], other non-urothelial malignant tumors, or simply a non-neoplastic lesion.

In this practical context, urologists and pathologists must bear in mind a long list of conditions that, eventually, may cause diagnostic problems. This narrative revisits some of them based on more

than 25 years of clinical practice. More specifically, the following entities are considered: urothelial papillomas (conventional (UP) and inverted (IUP)), nephrogenic adenoma (NA), polypoid cystitis (PC), fibroepithelial polyp (FP), prostatic-type polyp (PP), verumontanum cyst (VC), xanthogranulomatous inflammation (XI), reactive changes secondary to BCG instillations (BCGitis), schistosomiasis (SC), keratinizing desquamative squamous metaplasia (KSM), post-radiation changes (PRC), vaginal-type metaplasia (VM), endocervicosis (EC)/endometriosis (EM) (müllerianosis), malakoplakia (MK), florid von Brunn nest proliferation (VB), cystitis/ureteritis cystica (CC), and glandularis (CG), among others. Cellular proliferations with concerning histological features and poorly understood etiopathogenesis like IgG4-related disease (IGG4), PEComa (PEC), and pseudosarcomatous myofibroblastic proliferations (post-operative spindle cell nodule (POS), inflammatory myofibroblastic tumor (IMT)) are also included in this review.

This review collects a list of well-documented examples of mimickers of urothelial carcinoma (UC) obtained strictly from the personal experience of the authors.

## 2. Urothelial Papilloma (UP)

UP is a benign condition classically described in pathology textbooks. It tends to occur in younger people than urothelial carcinoma, even children, and usually are solitary lesions. The World Health Organization (WHO) and the International Society of Urological Pathology (ISUP) established, in 1998, a consensus of its strict definitory criteria [2] since this limit was a permanent source of diagnostic misinterpretations. UP has a very low rate of malignant transformation or recurrence, as has been reported in a recent long term study of 41 patients [3].

Specifically, UPs are strict exophytic papillary lesions with delicate stalks, and sometimes edematous stroma, covered by a normal-in-thickness multilayer coat of benign-appearing transitional cells (Figure 1a). The histological distinction of UP from low-grade, exophytic intraepithelial urothelial carcinoma may be difficult in selected cases.

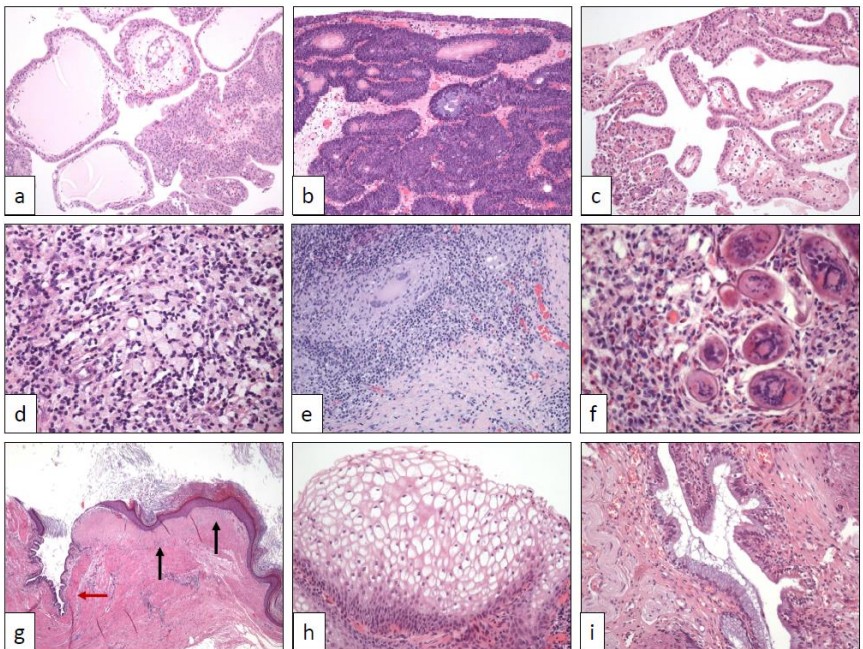

**Figure 1.** (**a**) Urothelial papilloma (40×), (**b**) inverted urothelial papilloma (100×), (**c**) nephrogenic adenoma (100×), (**d**) xanthogranulomatous inflammation (250×), (**e**) post-BCG inflammatory changes (100×), (**f**) schistosomiasis (400×), (**g**) keratinizing desquamative squamous metaplasia of the ureter (squamous metaplastic epithelium (black arrows) are mixed with normal urothelium (red arrow)) (20×), (**h**) vaginal-type squamous metaplasia (250×), (**i**) endocervicosis (250×).

Some molecular similarities have been described between UP and its inverted counterpart (see below).

## 3. Inverted Urothelial Papilloma (IUP)

The pathological diagnosis of inverted neoplasms remains a challenging issue in the urinary tract, as highlighted very recently [4]. The distinction between IUP and low-grade urothelial carcinoma with inverted growth is difficult and particularly important for patients. IUP is a classically recognized entity characterized by a benign clinical course in which recurrences are related to incomplete resections [5]. Although initially related with human papillomavirus, the last findings using immunohistochemistry and different FISH probes exclude this causal link [6].

The diagnosis of IUP is subjected to very strict morphological criteria [7–9]. IUP appears as slightly elevated lesions, mainly in the bladder trigone, on cystoscopy. However, some cases have been also described in the upper urinary tract as pedunculated (polypoid) masses [10]. Although the surface is usually flat and smooth, occasional papillary projections can be detected. Wavy small urothelial nests of cells without atypia displaying peripheral palisading immersed in a stroma without inflammation or fibrosis, and without invasion beyond the submucosa is the characteristic hallmark in IUP (Figure 1b). By contrast, large, rigid, and round urothelial nests with invasion into the muscular propria layer and accompanied by reactive stroma and/or inflammatory infiltrates strongly favor the diagnosis of urothelial carcinoma with an inverted growth pattern.

Several molecular studies have been performed in IUP [11,12]. Interestingly, IUP share with UP the RAS pathway activation in its genesis without *FGFR3*, *TERT*, *TP53*, or *RB1* gene implications, otherwise typical in urothelial carcinomas [13,14].

## 4. Nephrogenic Adenoma (NA)

NA is the resulting lesion of tubular renal cell desquamation seeding downstream along the urinary tract, from the renal pelvis to the urethra, as demonstrated some time ago by a molecular study of the lesion in a renal transplant-receiving patient [15]. The largest series of NA published so far [16] illustrates the varied histology of this condition and confirms its proven malignant simulator potentiality. NA tends to develop in the previously damaged urothelium, e.g., post-transurethral resections, urinary lithiasis, chronic inflammations, bladder catheterization, and any other situation creating the predisposing local conditions for the successful nesting of the physiologically desquamated tubular cells of the kidney.

The histological spectrum of nephrogenic adenomas is wide [17], with microcystic, cystic, papillary (Figure 1c), solid, and flat architectures, as well as clear cell, oncocytic, and fibro-myxoid changes. Pseudo-malignant features like muscle pseudo-invasion, nuclear atypia, and occasional GATA-3 positivity may occur [17]. Special attention should be paid to distinguish urothelial carcinomas mimicking nephrogenic adenomas, especially when NA present a pseudo-infiltrative growth pattern [1,17]. The combined expression of PAX-8 and absence of p63 and GATA-3 should solve difficult cases [16]. Very recently, napsin A has been proposed as a sensitive marker in NA useful as well in the differential diagnosis [18].

## 5. Polypoid Cystitis (PC), Fibroepithelial polyp (FP), Prostatic Polyp (PP), and Verumontanum Cyst (VC)

Chronic injury (lithiasis, catheterization, etc.) seems to be the main cause for developing the so-called PC [9]. Urologists are aware of this classic condition that, however, in some clinical contexts may mimic papillary urothelial carcinoma [19]. Histologically, large papillae with edematous stroma, inflammatory infiltrates, and narrow base is typically seen. The overlying urothelium may show hyperplastic changes but true dysplasia is lacking.

FP are non-neoplastic lesions that are most commonly seen in neonates and the upper urinary tract. Although they may reach a big size [20] and cause hematuria by torsion of the stalk, and other

urologic signs, a significant proportion of them is incidentally discovered in urologic explorations for other causes. The superficial urothelium may show hyperplastic changes and von Brunn nests, but true atypia is lacking.

PP and VC are occasional causes of lower urinary obstructive symptoms [21,22]. Interestingly, both conditions may mimic a prostate malignancy, particularly duct-type adenocarcinoma, since the periurethral location is the typical site where this prostate cancer subtype usually arises.

PP is made of irregular filiform stalk projections covered by benign prostatic epithelium. VC may display diverse morphology depending on the cyst origin. Usually, they are simple cysts covered by the epithelium of the prostatic periurethral glands including *corpora amilacea*, but NA and müllerian remnants may also present as cystic lesions.

## 6. Xantho-Granulomatous Inflammation (XI)

XI is a classical histological diagnosis in urinary and digestive tracts appearing typically as tumor masses on radiological studies. XI tends to fistulize to neighbor organs or to the peritoneum. The renal pelvis [23] is the most common location in the urinary tract, but cases arising in the ureter [24], bladder [25–27], and urethra [28] have been occasionally reported. Several microorganisms, especially *Proteus mirabilis* and *Escherichia coli,* have been implicated in its genesis generally associated with infective phosphate lithiasis, especially [29]. Although this condition mimics a tumor, its diagnosis does not preclude malignancy since cases of XI associated with cancer have been reported also in the urinary tract [25,30].

Grossly, XI also may simulate a neoplasm. Cut surface shows a destructive lesion composed of heterogeneous nodules with focal necrosis and whitish firm areas. Histologically, mixed inflammatory infiltrates including lymphocytes, plasma cells, and foamy histiocytes in variable proportions is the hallmark of XI (Figure 1d). Necrosis, when present, show a geographic appearance and includes polymorphonuclears and macrophages.

## 7. Bacillus Calmette-Guerin-Induced Inflammatory Reaction (BCGitis)

Intravesical instillations of BCG (*Mycobacterium bovis* strains) is a common urological practice in patients who have received a previous transurethral resection (TUR) for high-grade urothelial carcinoma (UC). The goal of this practice is to activate local host defenses as an inflammatory antitumoral instrument. The intimate mechanism of such immune reaction is a matter of study in the last years [31].

However, the florid inflammation caused is endoscopically indistinguishable from a neoplastic lesion. A TUR biopsy usually resolves the dilemma, since the typical tuberculoid granulomas (Figure 1e) are easily seen in the context of a chronic inflammatory infiltrate. A diagnosis of BCGitis does not preclude the presence of a concomitant UC recurrence or persistent carcinoma in situ.

## 8. Schistosomiasis (SC)

Human SC (snail fever) is a chronic parasitic disease produced by the genus *Schistosoma*, trematode flukes whose biological cycle alternate humans and *Bulinus*, a small tropical freshwater snail, as hosts. *Schistosoma haematobium*, one of the five species infecting humans, typically affects the urinary tract, most commonly the urinary bladder, where the trapped eggs induce chronic inflammation manifesting as hematuria and scarring [32].

The relationship between SC and bladder carcinogenesis has been largely reported. The local immune response produced by *Schistosoma hematobium* seems to be mediated by IL-4 signaling, which is responsible for the urothelial hyper-diploid hyperplasia occurring in this infection, which seems to be a key precursor of bladder carcinogenesis in the form of squamous cell carcinoma [33].

This disease is rare in non-endemic areas [32], but it is occasionally detected in non-endemic as a result of migratory movements from/to endemic countries. Ultimately, chronic non-treated infection

promotes an increased risk for squamous cell carcinoma development, particularly in Egypt and neighbor sub-Saharan countries, where this infection is endemic and this neoplasm more frequent.

An exuberant chronic inflammation with abundant eosinophils is detected on the histological analysis. Urothelium shows hyperplastic changes, but cytologic atypia is not a prominent feature. Depending on the stage of the disease and the extent of the biopsy specimens, a varied quantity of parasitic eggs, either calcified or not, can be seen in between a dense inflammatory infiltrate (Figure 1f), making the diagnosis evident. However, the histological diagnosis is impossible if parasitic eggs are not present in the sample. Here, serial sections of the submitted specimen should be performed if the clinical history and the histological picture is consistent with the diagnosis.

## 9. Keratinizing Desquamative Squamous Metaplasia (KSM)

KSM is a benign chronic condition affecting mainly the upper urinary tract, although the urinary bladder location has also been reported [34]. Recurrent episodes of lithiasis and urinary infections [35–37], as well as antecedents of renal tuberculosis [38] have been recorded, but the exact mechanism has still not been elucidated. A conservative therapeutic attitude has been proposed in these cases [36,37]. Bilateral synchronic cases have also occasionally been reported [38]. In selected cases, the possibility of urothelial carcinoma may be considered in the differential diagnosis on CT scans [35].

Histologically, the normal urothelium is replaced by well-differentiated, benign-appearing, squamous epithelium with prominent keratinization desquamating into the urinary tract lumen (Figure 1g).

## 10. Post-Radiation Changes (PRC)

Despite the significant advances obtained in radiotherapy, the bladder is frequently affected secondarily when cancers arising in neighbor organs, for example, prostate, rectum, and uterus, are treated with this kind of therapy [39]. These changes are more frequently seen within the first two years after radiation and present as episodes of intermittent hematuria.

PRC may be puzzling and confounding for the pathologist when analyzing small biopsies or transurethral resection specimens. Histologically, the urothelium shows hyperplastic pseudo-carcinomatous changes. There is increased cellularity with focal urothelial atypia consisting of nuclear pleomorphism and prominent nucleoli. On the surface, a papillary or pseudopapillary architecture may appear. Urothelial nests may also be seen in the lamina propria showing a pseudo infiltrative pattern. True atypia and mitoses, however, are not evidenced. The stroma is hyperemic and edematous, and some giant multinucleated cells can be found.

## 11. Vaginal-Type Squamous Metaplasia (VM)

VM is a frequent finding in women's bladder. Several decades ago, a large autopsy study showed that the trigone is the preferred location for this condition [40]. The term "pseudomembranous trigonitis" sometimes applied to this particular metaplasia associated with the urethral syndrome seems to be a misnomer [41,42]. On cystoscopy, VM is identified as whitish plaques in the urothelium. Such whitish plaques can be detected also in male patients [43], some of them receiving estrogens for prostate cancer [41]. Similar findings have been described in other clinical contexts, for example, in patients with a long history of bladder catheterization or secondary to chronic injury caused by lithiasis or chronic trigonal inflammation. Even more frequent VM can be the endoscopic hallmark of urethral pain in patients suffering cystitis-like symptoms. Some cases have been described in association with ureterocele raising the hypothesis of a heterotopic origin [44].

Such features may rise in the urologist the differential diagnosis between squamous metaplasia and even malignancy. The histopathological counterpart of such cystoscopy images corresponds to a particular type of squamous metaplasia in which cells appear enlarged with clear cytoplasm resembling the squamous epithelium of the vagina (Figure 1h).

## 12. Endocervicosis (EC)/Endometriosis (EM) (Müllerianosis)

EC and endometriosis refer to the presence of endocervical and endometrial tissue outside the endocervix and endometrium, respectively. The conceptual name müllerianosis merges these two terms with endosalpingiosis [45]. The urinary tract, especially the bladder, is the most frequently reported site for these conditions [46]. Bladder EC and endometriosis rise serious diagnostic concerns to clinicians and urologists since they usually appear as tumor masses on radiological exams (Figure 2a). Clinically, EM in the bladder presents in women with hypogastric pain, dysuria, and transient hematuria during menstruation, directly related to the hormonal cycle, which remains a key point for the correct diagnosis.

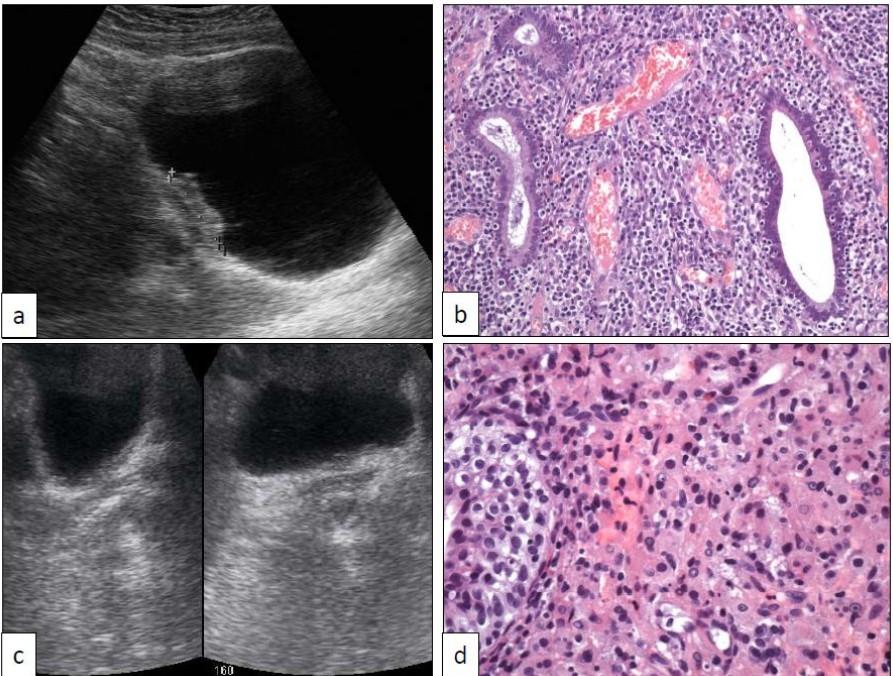

**Figure 2.** Bladder endometriosis ((**a**) sonographic and (**b**) histologic features (250×)) and malakoplakia ((**c**) sonographic and (**d**) histologic features (400×)).

Although benign, EC and EM may be recurrent and resistant to therapy. The capacity of malignant transformation is minimal but does exist at least from a theoretical viewpoint. In this sense, the first case of adenocarcinoma arising in bladder EC has been recently reported [47].

Both EC and EM are histologically well characterized [46,48,49]. Glands, either mucinous endocervical-type (Figure 1i) or endometrial (Figure 2a), appear within the bladder or ureteric wall with an infiltrative pattern accompanied by specialized stroma. This pseudo-infiltrative pattern may rise the erroneous diagnosis of adenocarcinoma [48]. Immunohistochemistry recapitulates the respective profile found in the endocervix [50] and endometrium.

## 13. Malakoplakia (MK)

MK is a subtype of chronic granulomatous inflammation that may involve different organs and systems, the urinary tract being one of the most commonly affected. Immunocompromised patients are typically affected. MK may reach occasionally a large size in the bladder closely mimicking a malignancy both radiologically (Figure 2b) and histologically [51]. However, the diagnosis of this condition does not preclude a concomitant diagnosis of urothelial carcinoma, as it has been recorded occasionally [52].

Histologically, MK is characterized by the presence of Michaelis-Gutmann bodies in a context of a chronic inflammation mainly composed of abundant histiocytes, the so-called von Hansemann's macrophages (Figure 2b).

## 14. Florid von Brunn Nest Proliferation (VB), Cystitis Cystica (CC), and Cystitis Glandularis (CG)

VB is very frequently seen all along the urinary tract, especially in the bladder and ureter. Sometimes they may cause diagnostic problems because they are numerous (florid VB) and display a pseudo-infiltrative pattern into the lamina propria, or because they suffer a prominent cystic transformation (cystitis/ureteritis cystica (CC)) (Figure 3), or because they develop extensive glandular mucinous metaplasia (cystitis/ureteritis glandularis) (CG).

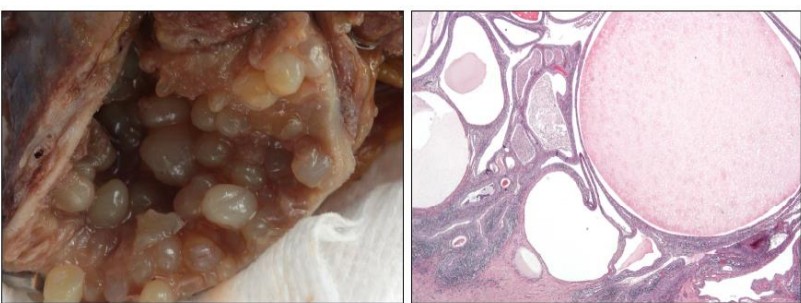

**Figure 3.** Gross and microscopic (20×) details in a florid case of ureteritis cystica.

Although they are classical entities well-known by pathologists, eventually they may pose diagnostic problems when the submitted material is scarce, superficial, or damaged by the resection procedures. For example, a florid VB must be distinguished from the *nested* variant of urothelial carcinoma [1]. This differential diagnosis is of crucial importance for the patient since *nested* urothelial carcinoma is an aggressive subtype of cancer. Mutations in *TERT* promoter has been proposed as a good discriminator between nested carcinoma and their mimickers [53], but this approach is not always available everywhere. In consequence, the dilemma may even be irresolvable in selected cases, and a new biopsy should be requested before making an overdiagnosis. In the same sense, CC must be distinguished from the microcystic variant of urothelial carcinoma [1]. Glandular mucinous metaplasia may occur in the epithelium of von Brunn nests. This change is usually focal and goblet cells are well differentiated, without atypia.

## 15. IgG4-Related Disease (IGG4)

IGG4 is a systemic autoimmune disorder that may affect many different organs and whose pathophysiological bases and intimate mechanisms are still badly known [54]. The urinary tract, from the renal pelvis to urethra [55–60], is frequently involved. IGG4 simulates malignancy everywhere since it develops tumor masses that are detectable on radiological exams. Radiologists have established differential diagnostic criteria to distinguish this disease from malignant neoplasms, for example, at the urinary tract [61].

The diagnostic criteria of IGG4 have been a matter of consensus among pathologists [62]. Since a wide number of inflammatory, neoplastic, and autoimmune diseases other than IGG4 do include dense plasma cell infiltration, the confident diagnosis of this disease must include the following three microscopic findings: dense lymphoplasmacytic infiltrate, storiform-type fibrosis, and obliterative phlebitis (Figure 4). Besides, IgG4 subpopulation among the total IgG plasma cell infiltrates must be higher than 40%. Phlebitis can be accompanied by thrombosis. Arteritis is not a feature.

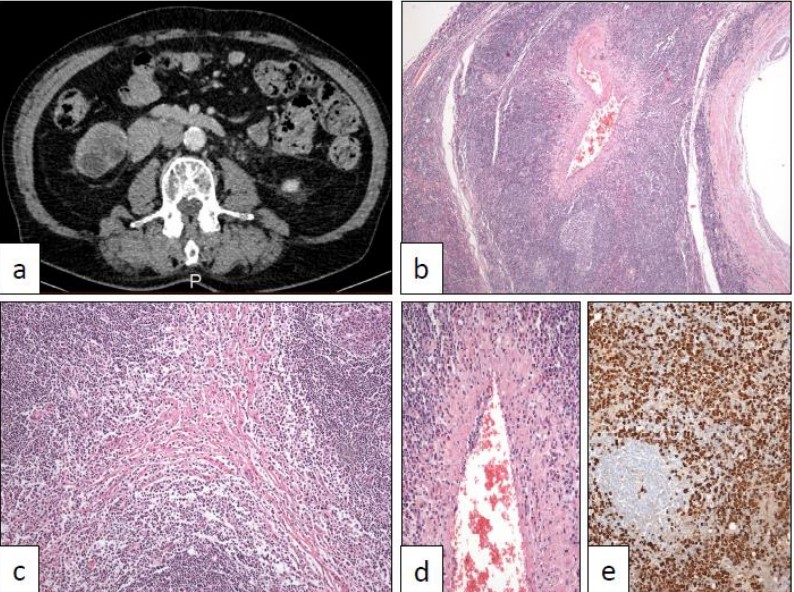

**Figure 4.** IgG4-related disease showing a tumor mass in the right ureter in CT scan (**a**) affecting ureter wall and periureteral soft tissues with a dense lymphoplasmacytic infiltrate (10×) (**b**) and displaying the typical storiform-type fibrosis (40×) (**c**), phlebitis (100×) (**d**), and dense IgG4 subpopulation higher than 40% (250×) (**e**).

## 16. PEComa (PEC)

PEC is an uncommon mesenchymal tumor very rarely reported in the urinary tract [63,64]. Allelic loss of the *TSC2* locus in 16p13 has been detected in PEC and angiomyolipomas, suggesting a common etiopathogenetic pathway of these entities [65], and, more interestingly, a hypothetical common therapeutic approach. Aside from the conventional histology, a sclerosing variant has been reported with a special predilection to be originated from for the retroperitoneal location [66]. Owing to its extraordinary rarity, PEC of the urinary tract can be preoperatively considered as urothelial carcinomas (Figure 5), so pathologists must be aware of this remote, although possible, diagnosis. Although some aggressive cases have been reported in the literature [67], most PEC pursues a benign clinical course.

Histologically, PEC appears as a proliferation of epithelioid cells arranged in solid nests and lobes (Figure 5). Proliferating cells show eosinophilic granular cytoplasm. Nuclear atypia and mitosis are not seen. Sclerosing PEC shows a slightly different microscopic appearance, with sheets of cells lying in a sclerosing stroma [66]. A faint melanocytic differentiation (HMB-45, Melan-A, etc.) (Figure 5), sometimes coupled with muscle markers expression (desmin, etc.), together with an absence of epithelial markers (cytokeratins, EMA, etc.), is the definitory immunohistochemical hallmark of this entity [64].

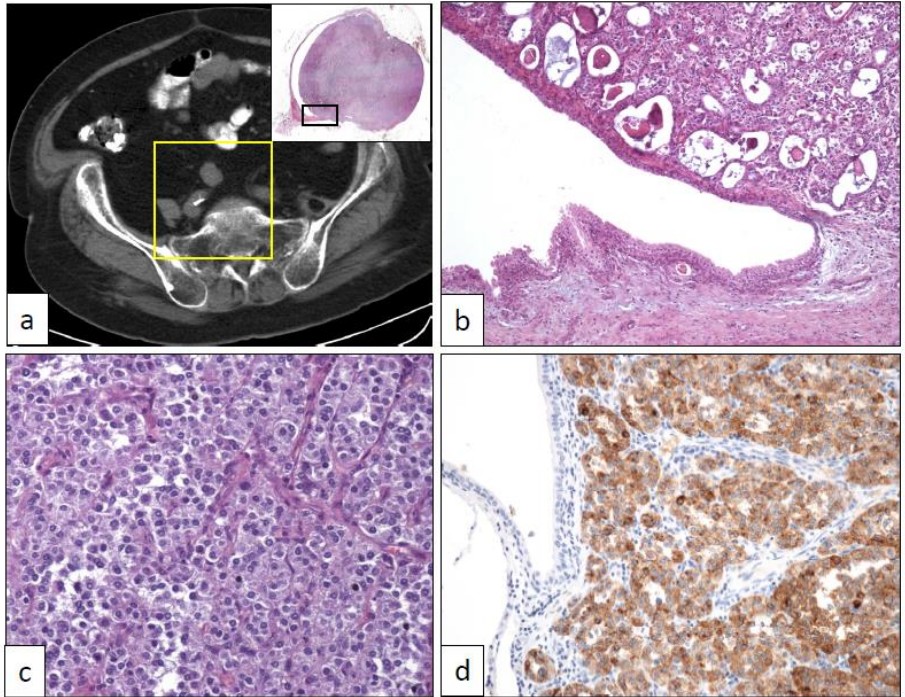

**Figure 5.** (**a**) Conventional PEComa arising in the right ureter (yellow square in the CT scan shows the panoramic view of a transverse section of the ureteral hourglass tumor in the upper right inset), (**b**) low-power view of the ureteral tumor corresponding to the rectangle black inset in (**a**) (40×), (**c**) medium-power view of the tumor showing solid cell nests with an epithelioid appearance (250×), (**d**) intense positivity with HMB-45 immunostaining (250×).

## 17. Pseudosarcomatous Myofibroblastic Proliferations

Myofibroblastic proliferations simulating malignant neoplasms are occasionally seen in the urinary tract, especially in the bladder. They are a classically ill-defined group of entities with a low grade of clinical recurrences and no metastatic potential [68], even after long-term follow-up [69]. Some secondary malignant transformations, however, have been occasionally recorded [70]. Their clinical, radiological, and histopathological features are always very threatening. At least two different conditions with poorly known etiopathogenetic mechanisms are included under this term: the postoperative spindle cell nodule (POS) and the inflammatory myofibroblastic tumor (IMT).

An exuberant stromal proliferation has been recognized a time ago in some patients who had received a previous transurethral resection for bladder cancer [71]. This "reactive" proliferation appears usually in the post-surgical radiological follow-up of the patients and raises immediately the suspicion of a tumor recurrence (Figure 6). Histologically, the lesion is also very concerning. The picture is dominated by a dense cellular spindle cell proliferation with marked atypia and mitosis (Figure 6). The main differential diagnosis of POS is a recurrent UC with sarcomatoid transformation. Despite its threatening appearance, the lesion is benign, being considered as a florid reparative overgrowth of fibroblasts in response to previous surgical injuries. Clues for its correct identification are the surgical antecedent, the absence of true tumor necrosis, and the usually exophytic and non-infiltrative growth pattern at the deep border within the bladder wall.

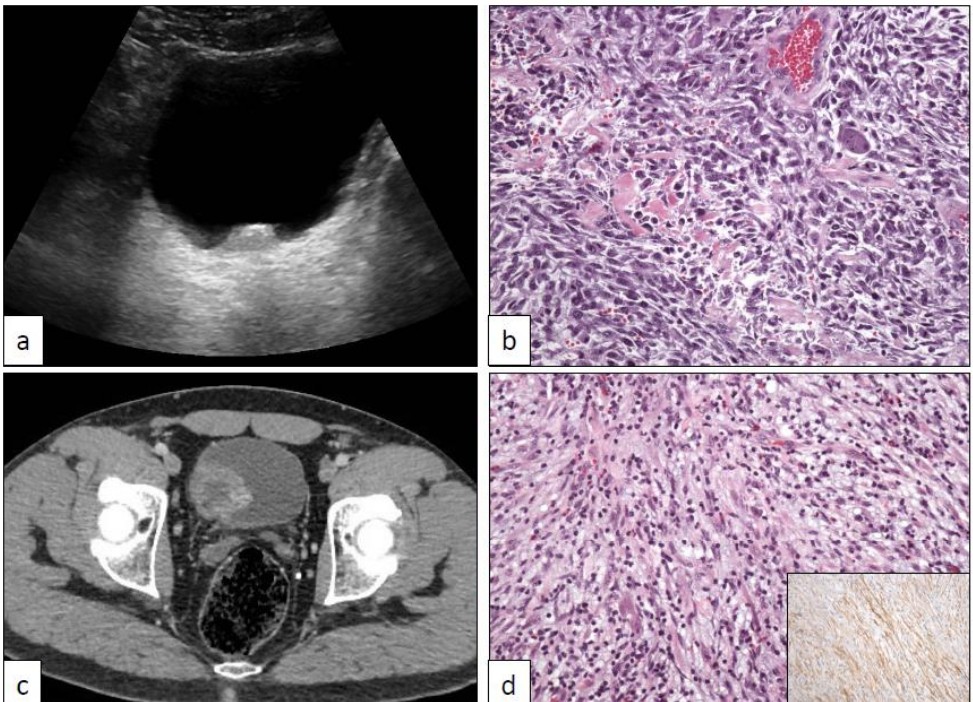

**Figure 6.** Post-operative spindle cell tumor of the bladder showing typical sonographic (**a**) and histological (**b**) features. Inflammatory myofibroblastic tumor of the bladder displaying characteristic CT scan (**c**) and histological (**d**) findings.

IMT in the urinary tract, same as in other locations, is a terminologically confusing entity that has received different names in the literature, such as pseudosarcomatous myofibroblastic proliferation or inflammatory pseudotumor. Radiologically it is also very concerning because it appears as large tumors (Figure 6) without previous history of transurethral resection. Histologically, a proliferation of loosely arranged spindle cells accompanied by inflammatory cells is the hallmark (Figure 6). Densely packaged areas may alternate with others with a myxoid-appearing background. ALK is positive in a subgroup of cases. Other immunohistochemical markers, however, do not provide useful definitory data. Some authors have compared IMT of the urinary tract with nodular fasciitis [72], however, a study has provided molecular evidence to distinguish them based on *USP6*, *ROS1*, and *ETV6* gene rearrangements [73]. More recently, another study has determined that these lesions are characterized by recurrent FN1-ALK fusions [74]. Finally, *TERT* promoter mutation may be of help distinguishing problematic spindle cell lesions of the urinary bladder [75].

## 18. Conclusions

The exact clinical context and the peculiarities of individual cases make the list of mimickers of urothelial carcinoma very long and varied, and this narrative intends to merge all of them in a readable review illustrated with a profusion of the typical pictures of most of the included entities. When possible, the difference in the urologist's and pathologist's approaches are specified, always focusing on the essential points. There are benign tumors, metaplastic and reactive changes, hyperplasias, pseudotumors, infections, and inflammatory conditions. The authors encourage the readers for a collaborative multidisciplinary work that will assure its correct recognition, avoiding overtreatments.

**Author Contributions:** C.M., J.C.A., and J.I.L. conceived, designed, and wrote the manuscript. All the figures are original and have been obtained from the authors' collection. All authors have read and agreed to the published version of the manuscript.

**Funding:** This study received no external funding.

**Institutional Review Board Statement:** Not applicable.

**Informed Consent Statement:** Not applicable.

**Conflicts of Interest:** The authors declare no conflict of interest.

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
