# Peer review of "Mimickers of Urothelial Carcinoma and the Approach to Differential Diagnosis"

_clinpract, doi:10.3390/clinpract11010017_

Round 1

Reviewer 1 Report

From a practical standpoint, a manuscript entitled "Simulators of urothelial carcinoma" is a valuable contribution to the literature relevant for the clinician. However, some changes have to be instituted before it can be accepted for publication. First and foremost, the level of English is substandard in many parts of the paper; therefore, extensive proofreading by a native English speaker is definitely warranted. Even in the title the word 'simulator' is not appropriate, I propose changing the title to "Mimickers of urothelial carcinoma and the approach to differential diagnosis". Furthermore, all abbreviations should be initially stated in full, especially in the Abstract section of the manuscript. This is also valid for microorganism names (e.g. Proteus mirabilis and Escherichia coli - line 127). All names of biological organisms have to be italicized (this includes the genus Bulinus, line 154). Some smaller paragraphs can be merged for clarity purposes (e.g. Polypoid cystitis, Fibroepithelial polyp and Prostatic-type polyp and verumontanum cyst). It is not clear whether all figures are original and from the authors themselves; hence, this should be noted. When intravesical instillations of BCG are mentioned, the procedure and the contents (Mycobacterium bovis strain) should be described. It should also be mentioned that, when pseudosarcomatous myofibroblastic proliferations are concerned, TERT promoter mutation analysis and detection of ALK expression/rearrangements are valuable additional diagnostic adjuncts, strongly supporting sarcomatoid urothelial carcinoma and PSMP, respectively. Please refer to Bertz et al. Histopathology 2020 Dec;77(6):949-962. doi: 10.1111/his.14206. Finally, in 'Author Contributions', it is mentioned that "the manuscript is based solely on the own experience". However, the researchers cite plenty of papers, so this designation is not completely fair.

Author Response

The authors thank the suggestions made by the referee. All of them have been taken into account and the changes added in red in the manuscript.

  1. The English has been improved throughout the manuscript
  2. The title has been changed as suggested
  3. Abbreviations have been included starting from the Abstract.
  4. Microorganisms names have been changed as proposed and italics have been used.
  5. Smaller paragraphs (polypoid cystitis, fibroepithelial polyp, prostatic-type polyp, verumontanum cyst) have been merged.
  6. All the figures along the paper are original from the authors. Although some entities have been previously published by the authors, the specific pictures for this review are all original and previously unpublished. This point has been noted in the authors’ contributions.
  7. Mycobacterium bovis train has been included in the BCGitis paragraph.
  8. TERT promoter mutation as a help in the distinction of spindle cell proliferations in the bladder, together with the reference proposed, has been metioned and the reference included (#75).
  9. “The manuscript is based solely in the experience of the authors” is misleading and has been removed in Authors Contributions section. 

Reviewer 2 Report

In this paper, the authors review the current evidence on simulators of urothelial carcioma. I believe that this manuscript deserves publication after some changes.

-       The manuscript is well structured and provides a clear overview of the topic. Although this review is not systematic, the authors should state the methods that they used to identify the studies discussed in the manuscript. A specific section should be added.

-      Introduction section: although the authors correctly included important papers in this setting throughout the paper, we believe a couple of studies should be cited within the introduction (doi: 10.1016/j.urology.2007.02.017.; doi: 10.3390/cancers12061449), only for a matter of consistency. We think it might be useful to introduce the topic of this interesting study. 

Author Response

The authors thank the suggestions made by the referee.

  1. The authors have included a sentence at the end of the introduction stating that the cases used in this review (including all the pictures) are from the personal experience of the authors and have been collected along 25 years of clinical practice.

The referee suggest to include a case report of a “nested variant” of urothelial carcinoma and an extensive review of the current strategies and novel therapeutic approaches for metastatic urothelial carcinoma in the introduction for a matter of consistency. The nested variant has been recently reviewed by the authors as a mimicker of von Brunn nests (reference #1 in the paper). We honestly believe that a single case report is not enough to support such consistency. We refer to the review of “the unusual faces of bladder cancer” recently published for such a purpose, which has been quoted as reference number 1 in the introduction. On the other hand, we believe that a paper reviewing the current strategies and the therapeutic approaches of urothelial carcinoma is out of the focus of a review of cancer mimickers. 

Round 2

Reviewer 1 Report

Thank you for incorporating all the required changes.

Reviewer 2 Report

We recommend Acceptance.